# Xanthan-Based Materials as a Platform for Heparin Delivery

**DOI:** 10.3390/molecules28062757

**Published:** 2023-03-18

**Authors:** Narcis Anghel, Irina Apostol, Maria Valentina Dinu, Cristina Daniela Dimitriu, Iuliana Spiridon, Liliana Verestiuc

**Affiliations:** 1“P. Poni” Institute of Macromolecular Chemistry, Grigore Ghica-Voda nr. 41A, 700487 Iasi, Romania; 2Faculty of Medicine, “Gr. T. Popa” University of Medicine and Pharmacy, Universitatii nr. 16, 700115 Iasi, Romania; 3Faculty of Medical Bioengineering, “Gr. T. Popa” University of Medicine and Pharmacy, Kogalniceanu nr. 9-13, 700454 Iasi, Romania

**Keywords:** xanthan, oleic acid, polyurethane, heparin, drug delivery

## Abstract

Heparin (Hep), with its anticoagulant activity, antiangiogenic and apoptotic effects, and growth factor binding, plays an important role in various biological processes. Formulations as drug delivery systems protect its biological activity, and limit the potential side effects of faulty administration. The objective of this study was to develop novel xanthan-based materials as a delivery carrier for heparin. The materials exhibited remarkable elastic behavior and toughness without any crack development within the network, which also support their application for tissue engineering. It was found that all materials possessed the ability to control the release of heparin, according to the Korsmeyer–Peppas release model. All Hep-containing materials caused significant exchanges of the activated partial thromboplastin time (aPTT) and prothrombin time (PT) parameters, indicating that formulated natural/natural synthetic polymeric networks conserved heparin’s biological activity and its ability to interrupt the blood coagulation cascade. The obtained results confirmed that developed materials could be carriers for the controlled release of heparin, with potential applications in topical administration.

## 1. Introduction

Drug delivery systems based on natural polymers became of high interest in the last decades, due to their biocompatibility and biodegradability, as well as resource availability. The presence of reactive groups on the natural polymers is a great advantage, because other functional groups could be introduced, offering new physical and chemical properties. Thus, polymer-based materials’ properties can be tailored toward specific functionalities and adapted for different applications. One of the most important applications is their use as a platform for controlled drug release.

Polysaccharides, which are mostly obtained from plants, animals, microbes, and algae, are macromolecules that contain glycosidic linkages. In addition to serving as an energy source for organisms, polysaccharides play a crucial role in the functioning of cell membranes and other biological processes. Numerous biological processes, including immune system control, anti-tumor activity, control of gut flora, antioxidation, etc., are carried out by polysaccharides [1]. Polysaccharides’ structural characteristics are directly correlated with their biological activities. The structure of polysaccharides contains a variety of functional groups, including hydroxyl, amino, and carboxylic acid groups [2]. Novel biological activities of polysaccharides can be created by altering their chemical structure or adding additional groups.

To deliver medications to the target site efficiently and control their pharmacokinetics, effectiveness, toxicity, immunogenicity, and biometrics, drug delivery systems employ a variety of multidisciplinary techniques. However, the rate of drug release, tissue selectivity, and drug stability in vivo are difficult to anticipate. Drug delivery systems are built with a variety of materials and chemicals policies to address these issues [3]. The creation of new intelligent drug delivery systems has a lot of potential, thanks to the growth of nanotechnology. Since the structural units of nanomaterials are often smaller than cells, they can exhibit unique functions and properties by way of their size, surface and interface effects, etc.

Porous materials have gained significant attention in the field of drug delivery, due to their ability to control the release of drugs. Among various porous materials, xanthan, alginate, and polyurethanes are widely used for controlled drug delivery.

The current study was dedicated to the development of novel xanthan-based materials as a delivery carrier for heparin.

Xanthan (Xn) is a heteropolysaccharide that is produced by *Xanthomonas campestris* bacteria through aerobic fermentation. Repeating units of D-glucose connected by β1-4 linkages, as well as a side chain of D-mannose and D-glucuronic acid are present in the Xn macromolecular chain. The primary uses of xanthan are linked to its exceptional properties, including its high viscosity at low concentrations; its high solubility in hot and cold water; its viscoelastic behavior; its resistance to enzymatic degradation, temperature, and salt solutions; its interaction with other polymers; and also because of its straightforward processing mechanism. The xanthan molecule’s unique helical structure gives it remarkable qualities including its pseudoplastic nature and quick recovery, which expands its applications in a variety of areas such as industrial applications, biomedical engineering, and agricultural fields. In the pharmaceutical sector, drug delivery systems are where xanthan gum is most commonly used [4]. In both culinary and non-food applications, xanthan is primarily utilized as a suspending agent, viscosity maintainer, gel forming agent, and flocculator. The inclusion of xanthan to uniformly suspend solid components in formulations has improved flow behavior in fungicides, herbicides, and insecticides used in agriculture.

At low temperatures, Xn presents an ordered and rigid double helical strand structure while a disordered and flexible coil structure is recorded at high temperatures. This behavior, as well as its compatibility with metallic salts, stability under a wide range of pH levels, and high viscosity at low concentrations, make it useful in specific applications [5]. Xn is also used in healthcare applications as wound dressing material [6] or as delivery systems for biologically active compounds [7].

Numerous investigations have been conducted on hydrogels based on xanthan for drug delivery purposes. Xanthan hydrogels enable the controlled release of drugs. Modified Xn with succinic anhydride has been utilized to create hydrogels that are sensitive to ionic strength for the delivery of gentamicin. A swift release of gentamicin was initially observed as the ionic strength of the release medium increased. Subsequently, sustained drug release was observed due to the strong interaction between polymer molecules, creating a compact network [8]. An in situ hydrogel was synthesized to enhance the bioavailability and contact time between the drug and the precorneal membrane for drug delivery. Mucoadhesive polymers such as Xn and sodium alginate were employed, whereas poloxamer 407 and poloxamer 188 were used to encapsulate moxifloxacin hydrochloride. The formulations developed were uniform in consistency, transparent, with good spreadability, and optimal bioadhesion properties [9]. The pore size of xanthan-based materials can be controlled by adjusting the degree of crosslinking, which affects the diffusion of drugs through the material. For example, Lee et al. developed a xanthan-based hydrogel with tunable pore sizes for the controlled release of curcumin, an anti-inflammatory drug. The hydrogel showed sustained release of curcumin for up to 24 h, and the release rate could be controlled by varying the crosslinking density of the hydrogel [10].

Alginates (Alg) are natural anionic polysaccharides that present numerous advantages such as good availability, biocompatibility, and biodegradability [11,12], and low toxicity, thickening, and gelling properties, which make them suitable for numerous applications, including drug-controlled release systems [13,14,15]. Alginates consist of 1,4-linked β-d-mannuronic acid (M) and 1,4 α-l-guluronic acid (G) residues; their composition and the sequence of G and M residues are dependent on the sources used to extract alginate. It is worth mentioning that the M/G ratio and its structure substantially influence alginate properties. Alginates may also mix consistently with a range of other polymers to make films, membranes, microbeads, scaffolds, and hydrogels that work well as platforms for research into how cells interact with their microenvironments. It is important to remember that alginates could be formed in the form of gel under mild circumstances, unlike other polysaccharides [16]. Alginates do not affect the viability of cells, and are also safe for biological systems [17]. Due to their swelling properties through the hydrophilic groups, solubility (covalent bonds prevent them from dissolving even after swelling) [18], and pH-sensitive responsiveness, alginate-based polymeric systems have been shown to be more effective in controlled/sustained drug release investigations. Since alginates contain carboxyl groups, they aid in the dissolution of alginate in neutral and alkaline pH conditions, protecting the drug molecules from acidic environments. In contrast, microbial infections have been acknowledged as a severe issue in recent years, and the substance used should be harmless and widely available. As a result, the use of naturally occurring alginate as an antimicrobial agent and as a material for wound dressings has been growing quickly for years. Alginate-based porous materials have been extensively studied for drug delivery applications, due to their ability to form gels in the presence of divalent cations such as calcium ions. The gelation process forms a network of pores that can trap drugs and control their release. Alginate-based materials can also be easily modified by incorporating other polymers or molecules to enhance their drug delivery properties. For example, Balanč et al. developed porous alginate-based microspheres loaded with resveratrol, a natural antioxidant. The microbeads exhibited a sustained release of resveratrol for up to 8 h, and the release rate could be controlled by varying the concentration of calcium ions in the polymeric matrix [19].

Alginates are readily available in a variety of forms in moderate environments without the need of hazardous chemicals.

The synthetic polymers known as polyurethanes (PUs) have urethane (or carbamate) linkages (–NH–COO–) in their primary chains. The vast array of building elements that may be included into PUs allows for the customization of their material attributes to serve a number of purposes. It is significant that PUs can be produced in vast quantities and treated in a variety of ways. These factors have led to the widespread use of PUs in industry, such as for foams, coatings, fibers, adhesives, sealants, electronics, elastomers, actuators, and biomaterials [20,21].

Their biocompatibility, durability, and mechanical properties extended their application in the medical field to wound dressings, catheters, hospital beddings, or surgical drapes [22]. There has been a lot of interest in biodegradable PUs to address the requirement for tissue repair and regeneration. Biodegradable PU mimics the biomechanical behaviors of soft and elastic tissues by exhibiting high mechanical strength, softness, and high elasticity. The adaptable chemistry of PU synthesis can also produce a variety of biodegradable PUs to fulfill the unique requirements of various tissues. Hydrolyzable linkages, such as ester, amide, anhydride, and carbonate linkages, are incorporated into polymer backbones to create biodegradable polyurethanes (PUs) [23]. Thermoplastic polyurethanes (PUs) are linear, and can be produced into three-dimensional (3D) scaffolds using a number of manufacturing processes. They are referred to as elastomers because of their extreme elasticity and softness, which set them apart from widely used materials. Polyurethane-based porous materials are commonly used for drug delivery applications, due to their ability to form porous structures with high surface area and tunable pore size. The pore size of polyurethane-based materials can be controlled by adjusting the concentration of the porogen used during synthesis. The release rate of drugs from polyurethane-based materials can also be controlled by varying the degree of crosslinking, or by incorporating other polymers or molecules to modify the drug release behavior. For example, Wang et al. developed a polyurethane-based porous material loaded with doxorubicin, an anti-cancer drug. The material exhibited sustained release of doxorubicin for up to 24 h, and the release rate could be controlled by varying the degree of crosslinking [24].

It was reported [25] that PU exhibited overall biocompatibility and excellent resistance to thrombosis. Despite this, the development of polyurethane heart valves faces several points of stagnation, due to limitations in their long-term biological durability, as well as to the competition with mechanical and bioprosthetic valves. In this study, we used a water-soluble PU as a binder and plasticizer, in order to assure the mechanical strength of the developed materials [26]. This aspect is very important, while the mechanical integrity under dynamic flow conditions assures medical device function.

In conclusion, xanthan, alginate, and polyurethanes are versatile materials that can be used to develop controlled drug release systems with tunable properties. The structural importance of these materials lies in their ability to form porous structures with high surface area and tunable pore size, which can be used to trap drugs and control their release. These materials have shown great potential in drug delivery applications, and further research in this area is expected to lead to the development of more effective and efficient drug delivery systems.

Heparin (Hep) is a poly-anionic, poly-dispersive, highly sulfated linear polysaccharide consisting of alternating N acetyl-d-glucosamine and d-iduronate residues [27]. Heparin inhibits intracellular protein kinase activity during signal transduction [28], and modulates the function and expression of numerous growth factors. It has an important role in modulating smooth muscle cell migration, as well as in neointimal proliferation.

Hep interacts with proteins through hydrophobic effects and hydrogen bonding. Being water-soluble, it is present in some tissue such as arterial blood vessels, the liver, and the lungs. It is frequently used as an anticoagulant for the prevention of venous thrombosis and pulmonary embolism. Heparin binds to antithrombin III and accelerates the enzyme-neutralizing effect of a serine protease inhibitor, thus preventing the formation of thrombin. It was approached in our study because, in our opinion, the dosages and controlled release of heparin could expand its medical applications. Due to its low half-life, Hep is eliminated in a short time and, as a result, its frequent use can induce systemic side effects. The interest for materials that do not provoke thrombosis, that are associated with a high level of morbidity and mortality, is continuously increasing. This is why numerous efforts are directed towards improving the biomaterials used as devices in short- or long-term treatments to prevent thrombosis.

Having in mind that drug delivery systems allow reduced dosage and frequency of administration of drugs, thus reducing drug-related side effects, we used xanthan/modified xanthan, polyurethane, and alginate as a matrix to obtain materials with a controlled release of Hep. The materials were characterized in terms of mechanical and morphological properties, and the Hep release mechanism was monitored.

## 2. Results and Discussion

### 2.1. FTIR (Fourier Transform Infrared Spectroscopy) Analysis of Materials

FTIR spectra of the obtained materials are shown in Figure 1.

In Figure 1A are presented the FTIR spectra of xanthan–alginate-based materials. The broad adsorption bands at 3338 cm^−1^ are attributed to the presence of –OH groups. Stretching vibrations of aliphatic C–H were identified at 2923–2856 cm^−1^ [29]. The bands present in the region between 1633 and 1598 cm^−1^ were assigned to asymmetric and symmetric stretching vibrations of carboxylate ions. The presence of glucuronic acid residues from alginate were evidenced by the peak at 773 cm^−1^. The chemical modification of Xn was evidenced by the increase in the absorption band at 1737 cm^−1^, characteristic for the carbonyl group. Furthermore, an intensification of the stretching vibrations of the –CH_2_– groups at 2922 cm^−1^ was recorded [15,30]. In samples containing heparin (Xn–Alg–Hep and XnOA–Alg–Hep), bands at 1433–1434 cm^−1^ and 1029 cm^−1^ were recorded, corresponding to asymmetric stretching of –COO^-^ and to asymmetric stretching of C–O–C, respectively; meanwhile, in the region 1240–1155 cm^−1^, characteristic absorption bands of S=O asymmetric stretching associated to sulphate groups of Hep were observed [31].

The FTIR spectra of materials based on Xn and PU are presented in Figure 1B. It was observed that a broad signal between 3348 and 3373 cm^−1^ remained. This signal is due to the superimposition of –NH– stretching vibrations of the urethane group of the Pus, as well as to terminal Xn or XnOA groups. Furthermore, the presence of the carbonyl group from the PU structure was evidenced by the bands from the region of 1739–1625 cm^−1^ [32]. The presence of heparin in Xn–PU–Hep and XnOA–PU–Hep materials was confirmed by the appearance of signals between 1430 and 1404 cm^−1^ attributed to –COO^-^ group. The bands present in the region of 1037 cm^−1^ correspond to asymmetric stretching of C–O–C bonds [31]. In the region of 1249 cm^−1^, characteristic signals of asymmetric stretching of S=O group were observed.

Comparing heparin-containing samples’ spectra with the FTIR spectra of Hep, some shifting in the peaks was observed from 1434 cm^−1^ to 1431 cm^−1^, or from 1041 cm^−1^ to 1035 cm^−1^. These shifts were attributed to strong hydrogen bonding interactions between polymeric matrices and heparin [33].

### 2.2. Mechanical Properties

The uniaxial compressive measurements were used to assess the elasticity, toughness, and stability of various materials based on the xanthan/chemically modified xanthan matrix loaded with heparin (Hep). The assessment of the influence of the materials formulations on the compressive mechanical performance sponges revealed that all materials exhibited typical compressive stress–strain (σ–ε) profiles characteristic of macroporous materials, as shown in Figure 2A. All of the formulations could be compressed to over 50% strain without any fracture development, due to the complete release of solvent (ethanol) from the macroporous structures of the formulations (see SEM images, Figure 3) upon compression. In addition, as Figure 2A shows, the stress–strain profile of the Xn–Alg–Hep system comprised the following three domains: (1) up to 15% strain was observed as an initial linear elastic domain; (2) up to around 60% plateau domain strain was noted; and afterwards, (3) a densification domain as a result of the gradual compression of the pores was observed. On the other hand, the stress–strain profiles of the Xn–PU–Hep, XnOA–Hep, and XnOA–PU–Hep systems revealed only two regions: the linear elastic domain up to about 80%, 68%, and the 61% strain and beyond the densification domain. Therefore, the elasticity and toughness of the prepared materials could be controlled either by incorporation of PU within the Xn matrix, or by its modification with OA. Other authors reported similar results related to the same systems loaded with antifungal or anti-inflammatory compounds [15], or for other gels based on an alginate/gelatin methacryloyl interpenetrating network [34] nano clay and natural polysaccharides [35], or for sponge-based systems on two oppositely charged polyelectrolytes (chitosan and poly(cyclodextrin citrate)) [36].

The elastic moduli (G, kPa) of all formulations were determined from the slopes of the linear parts of the stress−strain curves (Figure 2B), in agreement with the protocol already established for materials having Xn as a matrix component [15]. As can be seen from Figure 2C, the Xn–Alg–Hep system exhibited the highest elastic modulus, being fifteen-times higher than that of the Xn–PU–Hep system.

The mechanical properties of the Hep-containing formulations depend on the composition of the formulation of the matrix, i.e., the presence of Alg or PU in the mixture with Xn (Figure 2C,D). All samples exhibited remarkable elastic behavior and toughness without any crack development within the network, which also supports their applications for tissue engineering. The Xn–Alg–Hep formulations exhibited a modulus of elasticity of 36.71 kPa and a compressive strength of 49.94 kPa at 77.34% strain, while the Xn–PU–Hep formulations displayed a modulus of elasticity of only 2.48 kPa and a compressive strength of 34.37 kPa at 80.44% strain, which support a robust network with high toughness for the former ones. The use of XnOA in the preparation of Hep-loaded formulations led to more rigid networks, since the maximum sustained compression decreased to 68.39% for the XnOA–Alg–Hep formulation and 60.86% for the XnOA–PU–Hep material. Thus, the sustained compression of Xn-based formulations could be modulated either by changing the entrapped polymer (Alg or PU) or by modification of Xn with AO. The variety of pore sizes displayed by the Xn-based formulations (Table 1) support different applications for these materials. Generally, large pores are beneficial for cell attachment [37], while small pores enhance the mechanical performance of porous scaffolds/constructs [38]. Thus, the improvement in the compressive nominal stress of materials could be correlated to the decrease in pore sizes and in the wall thickness (Table 1), and to non-covalent interactions established between components of materials, which provide the basis for energy dissipation required for the improvement in the material’s mechanical properties [39].

### 2.3. Drug Delivery

The release of drugs from the polymeric matrix was also studied. The importance of mathematical models in the evaluation of drug release processes is well known. The kinetics models have the role of clarifying the release mechanism, and allowing the measurement of some important parameters.

Two mathematical models were used to fit the release data: the Korsmeyer–Peppas and Higuchi models [40].

The Higuchi equation is based on Fick’s first law of diffusion, which was the starting point for quantitative measurements in the controlled release studies. The Higuchi model was developed to cover various porous systems and especially for evaluating the release kinetics of water-soluble and encapsulated materials with a low solubility that are encapsulated into solid matrices.

The Higuchi model was used to assess the release kinetics of the active agents from the porous materials. Equation (1) represents the simplified Higuchi model:(1)$$M_{t}=k_{H}\times \sqrt{t}$$
where k*_H_* is the release constant of Higuchi expressed in mg × min^−1/2^, and *M_t_* is the concentration at time t.

The Korsmeyer–Peppas model was developed to describe the drug release from polymeric systems (Equation (2)):(2)$$\frac{M_{t}}{M_{\infty }}=k\times t^{n}$$
where *M*_∞_ is the amount of drug in the initial state, *M_t_* is the amount of drug released at time t, k is the release rate constant expressed in min^−n^, and n (dimensionless) is the exponent of release as a function of time t.

A value of the exponent of release equal to 0.5 indicates a Fickian diffusion mechanism of the drug from the inside of the material (the drug release is governed by diffusion), while a value of the exponent of release between 0.5 and 1 indicates non-Fickian diffusion (the drug release is governed by the swelling or relaxation of the polymeric chain).

Hep was not covalently bound in either the Xn–Alg/XnOA–Alg or Xn–PU/XnOA–PU matrix, but there was no doubt that electrostatic synergy was formed between heparin and the matrix components. A similar release trend was observed for all materials.

Figure 4 and Table 2 show information related to the Hep release profile from the studied materials. According to this information, our experimental data were best fitted by the Korsmeyer–Peppas model.

Xn–Alg–Hep, XnOA–Alg–Hep, and XnOA–PU–Hep presented non-Fickian transport (n value between 0.63 and 0.69), with heparin being released due to the combined effect of diffusion and polymer swelling. The amount of drug released at the initial burst phase, as well as the cumulative amount of drug released, were related to the porosity of the microparticles.

As seen in Figure 4, the materials comprising unmodified Xn presented a faster release profile, probably due to its hydrolytic degradation (rate constant between 2.6–2.9).

The use of XnOA as a polymeric matrix component led to a slower release process of Hep (rate of the drug release between 0.5–1.7).

The values of the transport parameter n (Table 2), being greater than 0.5 for all the tested materials, indicated non-Fickian diffusion. However, a value of 0.88 suggests a more significant contribution of swelling and/or erosion to the drug release mechanism, compared to a value of 0.62. This means that the release rate may slow down more significantly over time for materials with a greater n value, due to the increased effect of swelling and/or erosion on the polymer matrix (Figure 4).

It is important to note that the value of n was not the only factor that determined the drug release rate and mechanism. Other factors, such as the chemical composition, structure, and degradation characteristics of the polymeric matrix, can also affect the drug release behavior.

The literature supports the notion that a smaller pore size distribution and lower porosity can lead to a slower release rate of drugs, while a larger pore size distribution and higher porosity can lead to a faster release rate of drugs. The specific relationship between pore size, porosity, and drug release rate can vary depending on the specific drug and delivery system, but these general trends have been observed in numerous studies.

One study investigated the effect of pore size and porosity on drug release from a polyurethane-based matrix system containing paclitaxel. The results showed that increasing the pore size and porosity of the matrix increased the drug release rate. This was attributed to the increased diffusivity of the drug through the larger pores, and the increased surface area of the matrix due to the higher porosity [41].

Another study evaluated the effect of pore size and porosity on the release of red cabbage’s anthocyans from an alginate cryogel. The study found that increasing the pore size and porosity of the cryogel increased the drug release rate. The authors attributed this to the increased penetration of the release medium into the cryogel and the higher surface area available for bioactive principle diffusion [42].

In a similar study, the effect of pore size and porosity on the release of doxorubicin hydrochloride from an alginate hydrogel was investigated. The study found that increasing the porosity of the hydrogel increased the drug release rate, while increasing the pore size had a minimal effect. The authors attributed the increase in drug release rate to the higher surface area available for drug diffusion in the more porous hydrogels [43].

Corroborating the data from Table 1 and Table 3 with those concerning the release rate of heparin (Figure 4), indeed, its release rate was higher in the case of the Xn–PU–Hep material, which had the highest porosity (83.11%) and pore size (49.39 µm).

As expected, the chemical modification of xanthan determined a greater degree of packing of the polymer chains by establishing hydrophobic interactions between the hydrocarbon chains of the oleic acid, which led to a decrease in the porosity and pore sizes of the materials in question. All of this is reflected by the decrease in the release rate of heparin in the environment (e.g., XnOA–Alg–Hep).

The biggest influence on the release speed was related mainly to the porosity of materials, with the pore size having a smaller affect on the delivery process.

Overall, this study suggests that there is a correlation between drug release from a xanthan/alginate and xanthan/polyurethane matrix, the porosity, and pore dimensions. However, the exact nature of this correlation may depend on the specific drug and matrix system being used.

### 2.4. Antithrombotic Activity

Blood–biomaterials interactions are dependent on various pathological/health biological parameters, and also are strongly connected with the surface properties and the end application of the medical devices [44]. Blood contains cells (platelets, red cells, and white cells) and plasma; the liquid phase is abundant in proteins and other small molecules, and the surface of the biomaterial interacts with these components. Blood–surface interactions are very complex and involve biological processes such as protein adsorption, fibrinolysis, complement activation, platelet interactions, blood coagulation, and other cellular reactions [45]. Activated partial thromboplastin time (aPTT) and prothrombin time (PT) are reliable blood tests used to evaluate the behavior of the material. The aPTT values provide information about the effect of tested materials on possible delays in blood coagulation through the intrinsic pathway and prothrombin time (PT) to assess the material-induced changes in the extrinsic pathway of the coagulation cascade [46,47]. The in vitro anticoagulant activity of the prepared materials was evaluated in coagulation assays, such as the aPTT and PT, and other coagulation parameters, as the end-point for heparin-induced antithrombotic activity. The obtained results for some coagulation factors are presented in Table 4. Despite the analyzed materials containing a small amount of heparin, all of them caused significant exchanges of the aPPT and PT parameters (prolonging of the aPTT and PT), and the values exceeded the measurement limit of the instrument, 600 s and 120 s, respectively, indicating an antithrombotic effect. Similar data have been reported by Liu et al. on silk fibroin–polyurethane film containing heparin, as blood compatibility materials [48].

It is generally acknowledged that plasma proteins are firstly adsorbed onto the surface of the material, and can initiate the coagulation cascade. Fibrinogen (Fg) plays a key role in the coagulation cascade, as the protein can bind platelet glycoprotein IIb/IIIa receptor and activated platelets. Low Fg values indicate blood compatibility of the material surface, and the normal biochemical domain in the blood is 200–400 mg/dL. All of the materials with Hep induced a decrease in fibrinogen values. Heparin is an indirect anticoagulant that activates antithrombin via a high-affinity pentasaccharide sequence, and promotes its capacity to inactivate thrombin and coagulation factor Xa. As a result of inactivation, factor Xa showed a reduced ability to bind to fibrinogen, and Hep-induced antithrombotic activity resulted [49]. The values for other proteins, albumin (g/dL), and total proteins (g/dL) were comparable to those of the reference. Generally, the hematocrit (the percentage of red blood cells in blood, the oxygen carrier) and hemoglobin levels were slightly increased in the presence of the tested materials, but the values for these biological parameters remained within their normal domains. Having in mind that the sustained release of heparin from different polymeric matrixes is still in its early stage of development, there are many several issues to be managed in order to reach clinical applications.

## 3. Materials and Methods

### 3.1. Materials

Xanthan gum (Xn), with a molar ratio of D-glucose:D-mannose:D-glucuronic:pyruvic acid ketal:O-acetyl of 3.0:3.0:2.0:0.6:1.7, and a molecular weight of approximately 2.5 × 10^6^ Da from CP Kelco, U.S., was used as matrix. Heparin sodium (Hep, >150 IU/mg) was acquired from Sigma-Aldrich and used as received. 4-toluenesulfonyl chloride (TsCl), pyridine (Py), methylene chloride, oleic acid, 4,4′-methylene dicyclohexyl diisocyanate (H12MDI), 4,4′-diphenylmethane diisocyanate (MDI), dimethylol propionic acid (DMPA), polyhexamethylene carbonate diol (PHC–M 2000), 1,4-butanediol (BD), and triethylamine (TEA) were purchased from Sigma-Aldrich (USA).

### 3.2. Synthesis of Xanthan Oleate

The esterification of xanthan with oleic acid was performed according to the method presented by Dimofte et al. [15]. The esterification of xanthan with oleic acid was performed in the presence of TsCl and Py, in methylene chloride. A round-bottomed flask was used, and varying amounts of TsCl (5.1 g), pyridine (6 mL), oleic acid (8.5 mL), and methylene chloride (50 mL) were mixed together at room temperature using a magnetic stirrer. After 24 h, 10 g of xanthan was introduced to the flask and stirred for an additional 3 h. The resulting product (XaAO) was filtered and sequentially washed with methylene chloride, water, and ethanol, before being dried at room temperature.

### 3.3. Synthesis of Polyurethane

The synthesis of polyurethane was carried out following the method presented in reference [12]. Initially, 4,4’-methylene dicyclohexyl diisocyanate (H12MDI—10.48 g) or 4,4′-diphenylmethane diisocyanate (MDI—10 g), and 2.64 g of dimethylol propionic acid (DMPA) were stirred in 30 g of acetone (99.5 wt% purity) as the solvent. Additionally, 2–3 drops of dibutyltin dilaurate (DBTL) were included as a catalyst. The mixture was homogenized under reflux conditions (56 °C, 2 h). In the next step, 28 g of polyhexamethylene carbonate diol (PHC–M 2000) was added and stirred for 30 min. Then, 0.54 g of 1,4-butanediol (BD) was included as a chain extender, and stirring was continued for an additional hour at 56 °C. Finally, the polycarbonate urethane, which contained carboxylic groups, was neutralized using 2 g of triethylamine (TEA, 99 wt% purity) for 30 min. The reaction was completed by slowly adding deionized water (30 g) over approximately 30 min, resulting in an anionic polyurethane water dispersion.

### 3.4. Preparation of Biomaterials

Equal amounts of xanthan (Xn) or modified xanthan (XnOA) and polyurethane (PU) or alginate (Alg) were mixed in distilled water at a material:water ratio of 1:100 and heated at 70 °C for 60 min. To this mixture, heparin was added at a matrix:heparin ratio of 1:0.25. The codes of the materials obtained by freeze–thawing cycles, followed by lyophilization, are as follows: Xn–Alg–Hep; XnAO–Alg–Hep; Xn–PU–Hep; XnAO–PU–Hep.

### 3.5. FTIR (Fourier Transform Infrared Spectroscopy) Analysis

The FTIR spectra of the materials were recorded using a Vertex 70 FTIR (Brüker, Karlsruhe, Germany) that was equipped with an ATR device (ZnSe crystal) with a 45-degree angle of incidence. The spectra were analyzed in the ranges of 4000–400 cm^−1^ and 4500–600 cm^−1^. For the measurements, an average of 64 scans and a spectral resolution of 2 cm^−1^ were used.

### 3.6. Scanning Electron Microscopy (SEM)

SEM images were obtained at a magnification of 200× using a VEGA TESCAN microscope (Tescan, Kohoutovice, Czech Republic), with an acceleration voltage of 20 kV, at room temperature, with a low-vacuum secondary electron detector.

### 3.7. Determination of Materials’ Porosities

The density of the studied materials ($\rho_{pb}$) was calculated using Equation (3):(3)$$\rho_{pb}=\frac{W_{i}}{V}=\frac{W_{i}}{L\times l\times h},\;g/{\mathrm{cm}}^{3}$$
where *W_i_* (g) represents the dry weight of the sample, *V* (cm^3^) is the volume of the studied material (equal with the product of length, *L* (cm), width, *l* (cm) and height, *h* (cm) of the sample).

The porosities of the studied materials were determined using the saturation method described by Long et al. [50]. Ethanol was used as the wetting fluid. Material samples with weights between 0.0100 and 0.182 g were immersed in ethanol. After 24 h, the samples were withdrawn, and the excess ethanol was removed using filter paper. All of the samples were re-weighed, and their porosities were calculated according to Equation (4):(4)$$Porosity=\frac{W_{f}-W_{i}}{\rho_{EtOH}\times V}\times 100,\;\%$$
where *W_f_* is the weight of the sample after immersion in ethanol, and *ρ_EtOH_* represents the density of ethanol.

### 3.8. Mechanical Tests

Ethanol-swollen samples, as plates of about 8–10 mm thickness, 10–12 mm width, and 5–7 mm height, were tested using a Shimadzu Testing Machine (EZ-LX/EZ-SX Series, Kyoto, Japan). An initial force of 0.1 N was applied before performing each measurement. The cross-head speed was 1 mm × min^−1^, while the applied force was fixed at 20 N. The compressive stresses (σ, kPa), strains (ε), and elastic moduli (G, kPa) were evaluated at room temperature following the procedure previously published in [51,52].

### 3.9. In Vitro Drug Release

An amount equivalent to 25 mg of heparin from each formulation was incubated at 37 °C in 25 mL of phosphate buffer solution (PBS, pH = 7). Aliquots of 4 mL were taken at 30 min intervals for a total duration of 240 min, and the absorbance intensities at 260 nm were measured using a Jenway 6405 spectrophotometer (Jenway Ltd., Dunmow, UK). The concentrations of the released drugs were evaluated based on the calibration curve of heparin. In order not to change the hydrodynamic profile, after each sampling, the release medium was supplemented with 4 mL of fresh PBS solution. The calibration curve for heparin was plotted using the absorbances of the standard solutions containing different concentrations of heparin in PBS (0.05, 0.1, 0.2, 0.5, and 1 mg/mL) at 260 nm. The experiments were performed in triplicate, and the standard deviations (SDs) were computed.

### 3.10. Blood Coagulation

The measurements of the activated partial thromboplastin time (aPTT), the prothrombin time (PT), and fibrinogen were carried out according to the routine procedure [53]. In brief, PT, aPTT, and fibrinogen were measured on integral blood with anticoagulant (aqueous sodium citrate 3.8% *w*/*v*; ratio 1/9 *v*/*v*). The materials were incubated with 5 mL of blood at room temperature, for 30 min, and then were separated by centrifugation (2500 rpm, 10 min). The biochemical parameters in blood plasma were determined using a semi-automatic Helena coagulometer with 2 channels, photo-optical technique coagulation systems and a PT-Fibrinogen kit (International Sensitivity Index (ISI) = 1.07). The control sample was considered the free integral blood.

## 4. Conclusions

Materials containing heparin with controlled release were obtained using xanthan/modified xanthan, polyurethane, and alginate as matrices. The mechanical properties of the Hep-containing formulations were found to depend on the composition of the matrix. All of the samples exhibited impressive elastic behavior and toughness without any crack development within the network, making them suitable for tissue engineering. The Xn–Alg–Hep formulations showed a modulus of elasticity of 36.71 kPa and a compressive strength of 49.94 kPa at 77.34% strain, while the Xn–PU–Hep formulations displayed a modulus of elasticity of only 2.48 kPa and a compressive strength of 34.37 kPa at 80.44% strain, indicating a robust network with high toughness for the former formulations. The use of XnOA in the preparation of Hep-loaded formulations led to more rigid networks. Additionally, the dense pore walls provided great structural support to the entire interconnected porous network, resulting in high elasticity, flexibility, and non-brittleness in comparison with the Xn-based formulations.

The release of heparin in Xn–Alg–Hep, XnOA–Alg–Hep, and XnOA–PU–Hep exhibited non-Fickian transport (n values between 0.43 and 0.85), due to the combined effect of diffusion and polymer swelling. The use of XnOA as a component of the polymeric matrix led to a slower release process of Hep (release rate between 0.5–0.7) compared to the matrix containing chemically unmodified xanthan (2.6–2.9). Porosity was the most significant factor that affected the release speed, with a considerable statistical difference in the release rate of heparin from the tested materials. The critical Fisher coefficient (0.0393) was much lower than the theoretical value (322.57). The analysis of the variation coefficients showed that the difference in pore size between the tested materials had a relatively small impact on the drug release speed (with a negative coefficient value of −0.1776).

In coagulation assays, such as the aPTT and PT, and other coagulation parameters, the in vitro anticoagulant activities of the prepared materials were evaluated as the endpoint for heparin-induced antithrombotic activity. All Hep-containing materials prolonged the aPTT and PT, proving their antithrombotic effects. These results indicate that formulated natural/natural synthetic polymeric networks maintained the biological activity of heparin and its ability to interrupt the blood coagulation cascade.

The results confirmed that the materials based on alginate, polyurethane, and modified/unmodified xanthan could serve as carriers for the controlled release of heparin, with potential applications in topical administration. Together with the preclinical results in animals, these findings should encourage investigation of strategies that target new thromboresistance materials.

## Figures and Tables

**Figure 1 molecules-28-02757-f001:**
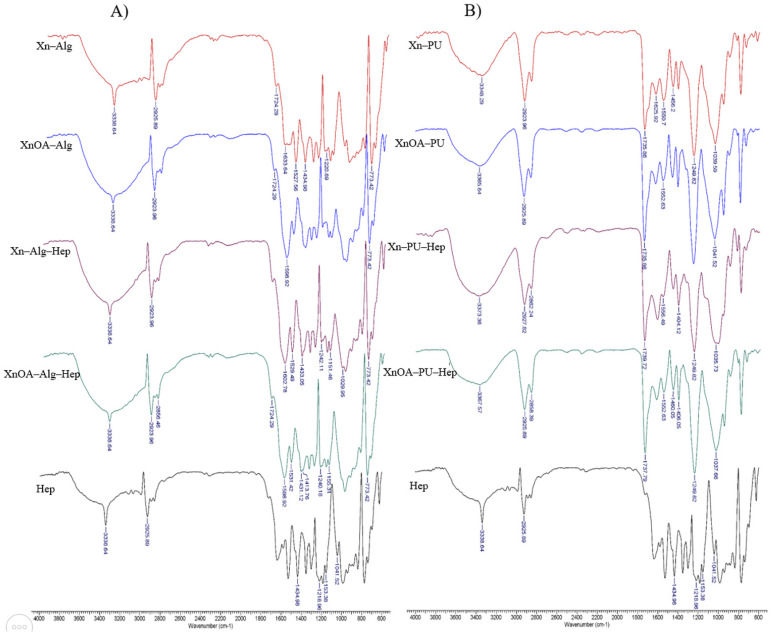
FTIR spectra of the materials ((**A**)—materials based on Xn–Alg matrix; (**B**)—materials based on Xn–PU matrix).

**Figure 2 molecules-28-02757-f002:**
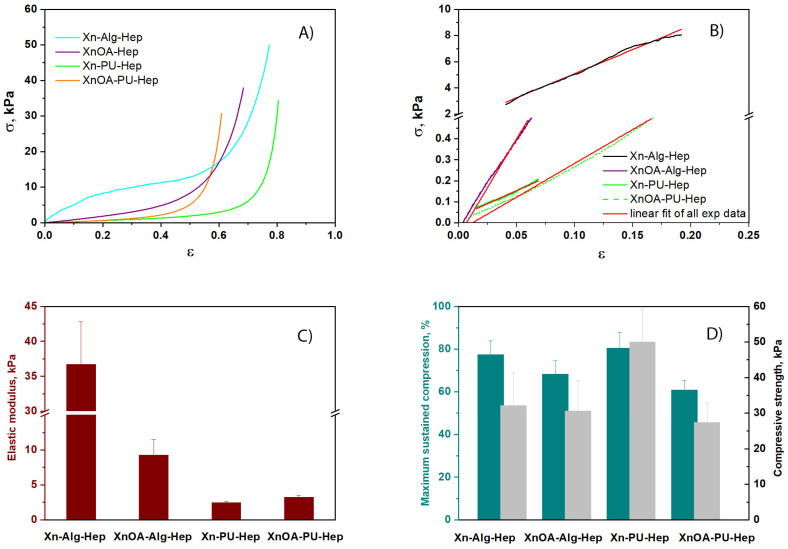
The mechanical properties of swollen Hep-loaded formulations under compression: (**A**) stress–strain profiles of Hep-loaded formulations; (**B**) linear dependence of stress–strain curves; (**C**) elastic modulus (wine color) determined according to the standard method; (**D**) maximum sustained compression (dark cyan color) and compressive strength (light grey color). The standard deviations are presented as error bars.

**Figure 3 molecules-28-02757-f003:**
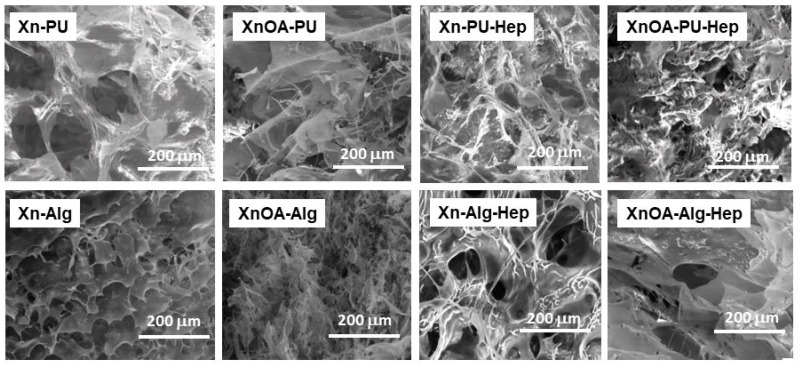
SEM micrographs of all materials.

**Figure 4 molecules-28-02757-f004:**
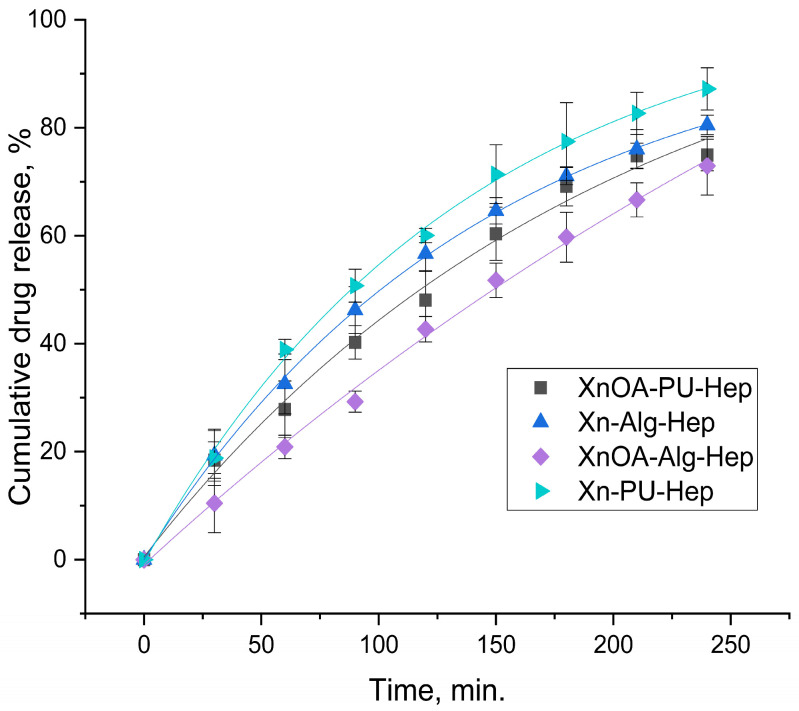
Cumulative drug release data for the tested materials according to the Korsmeyer–Peppas model.

**Table 1 molecules-28-02757-t001:** Average pore sizes and pore wall thicknesses for all sponges were determined from SEM micrographs with ImageJ1.41o.

Samples Code	Average Pore Size, μm	Average Pore Wall Thickness, μm
Xn–PU	110.03 ± 18.55	42.47 ± 4.79
XnOA–PU	50.35 ± 8.23	5.82 ± 1.31
Xn–PU–Hep	49.39 ± 9.94	11.24 ± 3.26
XnOA–PU–Hep	45.87 ± 15.50	30.42 ± 11.25
Xn–Alg	61.64 ± 17.68	90.79 ± 12.02
XnOA–Alg	53.28 ± 7.61	28.39 ± 6.28
Xn–Alg–Hep	46.66 ± 11.39	25.78 ±6.40
XnOA–Alg–Hep	45.73 ± 31.52	41.86 ± 4.14

**Table 2 molecules-28-02757-t002:** The kinetic parameters for drug release experiments.

Sample	Korsmeyer–Peppas Model			Higuchi Model	
	k, min^−n^	n	R^2^	k_*H*_, min^−1/2^	R^2^
Xn–Alg–Hep	2.6178	0.6318	0.99305	5.1032	0.97595
XnOA–Alg–Hep	0.5786	0.8879	0.99539	4.1623	0.90118
Xn–PU–Hep	2.9552	0.6249	0.99034	5.5620	0.97474
XnOA–PU–Hep	1.7456	0.6977	0.98959	4.7566	0.95603

**Table 3 molecules-28-02757-t003:** Density and porosity of the tested materials.

Material	Density, g/cm^3^	Porosity, %
Xn–Alg–Hep	0.021	71.26
Xn–PU–Hep	0.022	83.11
XnOA–Alg–Hep	0.024	44.43
XnOA–PU–Hep	0.030	60.0

**Table 4 molecules-28-02757-t004:** Preanalytical blood parameters for the characterization of antithrombotic activity.

Material	Hematocrit (%)	Hemoglobin (g/L)	PT (s)	aPTT (s)	Fibrinogen (mg/dL)	Albumin (g/dL)	Total Proteins (g/dL)
Reference	34.5	11.2	12.1	24.3	346	4.0	6.17
Xn–Alg–Hep	n.a.	n.a.	n.c.	n.c.	184	3.8	6.30
XnOA–Alg–Hep	46.8	18.7	n.c.	n.c.	296	4.8	6.56
Xn–PU–Hep	44.1	14.1	n.c.	n.c.	205	3.7	6.41
XnOA–PU–Hep	44.8	14.3	n.c.	n.c.	202	3.8	6.40

n.c.—no coagulation; n.a.—not available.

## Data Availability

The data that support the findings of this study are available from the corresponding author upon reasonable request.

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
