# Peer review of "Xanthan-Based Materials as a Platform for Heparin Delivery"

_molecules, 2023, doi:10.3390/molecules28062757_

Round 1
Reviewer 1 Report
· In the methodology part, in the synthesis of modified xanthan and polyurethane, it is more preferable to write the method then cite the reference
· Also, full details in the porosity determination should be added
· In section 3.7, line 319, please check the symbol of the porosity
· In section 3.9, full details for release study should be added, such as the volume, rpm, and the apparatus used.
· In section 2.2, line 199, please replace releasestudies with release studies
· The author did the FTIR for the platform and no data was shown. Please add the FTIR figure with discussion

Reviewer 2 Report
The current manuscript by Anghel et al. presents a short experimental study on the preparation of heparin-loaded xanthan gels, which were freeze-dried and used for the assessment of the release kinetics of the heparin. Outside of the scope of the study, the authors measured the mechanical properties of the gels after freeze-drying and the morphology of the gels (via SEM). As a result of the study, the authors reach trivial conclusions, such as “the mechanical properties of the Hep-containing formulations are depending on the composition of the matrix” (followed by a two-paragraph re-write of the empiric measurements) and “heparin-containing materials caused the prolonging of coagulation assays (aPPT and PT), proving their antithrombic effect”.
Although the study may contain new results on antithrombic materials preparation, it is poorly presented. Both conclusions are expected (trivial) and hardly original, the mechanical characterization if not related to the performance of the material in any way mentioned, the introduction presents general data for the chemistry of the precursors rather than information on their use as material builders or the optimal structure of the materials (and its relation to drug delivery). Further on, the methods section contains irrelevant information (use of FTIR) that has not been used in the study, whereas the pore size measurements that are done are not explained in a reproducible manner. I recommend the manuscript is rejected from publication in its current form.
I recommend the authors rewrite the introduction to explain the use of porous materials and their structural importance, then explain in detail the structural characterization and correlate it to the release kinetics (e.g. the difference in the "n").
Round 2
Reviewer 2 Report
The revised manuscript by Anghel et al. presents a short experimental study on the preparation of heparin-loaded xanthan gels, which were freeze-dried and used for the assessment of the release kinetics of the heparin. In the revised version of the manuscript, the authors addressed all my comments – they have emphasized the importance of the matrix structure by including an extended references list (consisting of new 15 publications), and by creating a link between the measured structural properties and the release kinetics. They have also added the missing FTIR data from the previous version of the manuscript, as well as improved the materials and method sections to be reproducible by others. I recommend the manuscript for publication as it is.
Author Response
Response to reviewer’s 2 comments for manuscript molecules-2200629 entitled
“Xanthan-based materials as a platform for heparin delivery”
We would like to thank you for taking the necessary time and effort to review the manuscript. We sincerely appreciate all your valuable comments and suggestions, which helped us in improving the quality of the manuscript.